# The Effects of Information and Communication Technology (ICT) Use on Human Development—A Macroeconomic Approach

Nada Karaman Aksentijević, Zoran Ježić  and Petra Adelajda Zaninović *

Faculty of Economics and Business, University of Rijeka, 51000 Rijeka, Croatia;
nada.karaman.aksentijevic@efri.hr (N.K.A.); zoran.jezic@efri.hr (Z.J.)
* Correspondence: petra.adelajda.zaninovic@efri.hr; Tel.: +385-51-355137

**Abstract:** Information and communication technology (ICT) is considered a significant factor in economic growth and development. Over the past two decades, scholars have studied the impact of ICT on economic growth, but there has been little research that has addressed the impact of ICT on human development, which is considered one of the fundamental factors of economic development. This could be especially important from the perspective of developing countries, which can develop faster through the implementation of ICT. Thus, the aim of this paper is to investigate the effects of ICT use on human development, distinguishing effects among high, upper-middle, lower-middle and low-income countries following the World Bank classification 2020. Our sample includes 130 countries in the period from 2007 to 2019. The empirical analysis is based on dynamic panel data regression analysis. We use Generalized Method of Moments (GMM) as an estimator, i.e., two-step system GMM. The results primarily support the dynamic behaviour of human development. The results of the analysis also show that ICT has highly significant positive effects on human development in lower-middle-income and low-income countries, while the effects do not appear to be significant in high- and middle-income countries. This research serves as an argument for the need to invest in ICT and its implementation in low-income countries; however, it also suggests that the story is not one-sided and that there are possible negative effects of ICT use on human development. From the perspective of economic policy, the results can be a guideline for the implementation and use of ICT in developing countries, which could lead to economic growth and development and thus better quality of life. On the other hand, policymakers in developed countries cannot rely on ICT alone; they should also consider other technological innovations that could ensure a better quality of life.

**Keywords:** human development; HDI; ICT; economic growth and development; dynamic panel analysis; GMM

## 1. Introduction

The development of technology, especially information and communication technology (hereinafter: ICT), has had significant effects on the economy and other aspects of human life in recent decades. It is impossible to imagine the effective functioning of an individual, an economy, and a whole society without the use of ICT. These effects are especially noticeable in the period of dramatic changes associated with the COVID-19 pandemic, when ICT allowed the "new normal" (Mińska-Struzik and Jankowska 2021; Huateng et al. 2021) to function. Educational institutions, the health system, enterprises, households, and the entire world economy depend on ICT. In the 1980s, in endogenous growth theories, scientists claimed that technological changes were the foundation of economic growth and stressed the importance of investing in human capital (Romer 1986; Lucas 1988; Grossman and Helpman 1991). In addition, according to the human development theory, income is only one of the elements that lead to the fulfilment of human needs; ICT is considered to have significant effects on the fulfilment of human needs, even greater than monetary income, because it improves the overall quality of life. ICT provides individuals with access

to information, enables social interaction, facilitates access to education and healthcare, and creates new business opportunities. Thus, ICTs can have both direct and indirect effects on the quality of human life. Furthermore, according to the European Parliament (2001) study, ICT offers remarkable opportunities to adequately address poverty in developing countries. Namely, ICT can assist the poor in business development or promote self-determination. From an equality perspective, ICT can also facilitate access to education and health, ensuring inclusion. Amartya Sen (2010) claims that information technology is responsible for the expansion of human freedoms and that it leads to better efficiency in various human activities. ICT is expected to have an even greater impact in the future. This impact is widely seen as positive, especially at the macro level as mentioned above, but at the micro level there are also possible negative consequences. From a psychological point of view, ICT also causes stress and anxiety, leading to modern diseases in developed countries. Developed countries are also expected to reach a steady state with respect to ICT, while developing countries are not; thus, ICT may have a greater impact in these countries. Therefore, ICT is an important topic in academic literature and public discourse. The objective of this study is to investigate and quantify the impact of ICT use on human development at the aggregate, i.e., macroeconomic, level. The motivation for this lies in the fact that in the macroeconomic view of development, the exclusion of ICT widens the gap between developed and developing countries (European Parliament 2001). We sought to investigate whether this statement is true and whether ICT plays a role in less developed countries. We argue that ICT can bridge the gap and make countries converge on the mean of human development. The main research hypothesis in this paper is that ICT has a significant impact on human development, but this varies according to the level of development of the country. The effects of ICT use on human development are analysed in high, upper-middle, lower-middle, and low-income countries, in accordance with the new World Bank classification of 2020. The motivation to differentiate ICT effects between countries with different levels of development also lies in the fact that most countries in the world are developing countries. According to the World Bank (2021), by income, there are 27 (12.45%) low-income economies, 55 (25.35%) lower-middle-income economies, 55 (25.35%) upper-middle-income economies and 80 (36.85%) high-income economies. This shows that most of the economies in the world are developing countries (63.15%). Therefore, this research highlights the fact that the most positive influence is on these countries. The main variable of interest is the Human Development Index (HDI), which represents a quality measure of human development (Karaman Aksentijević and Ježić 2018) and is used in the theoretical and empirical literature, which is detailed in the second part of the paper—the literature review.

This paper confirms previous findings of the importance of ICT in human development in developing countries. The results highlight the important role of ICT in lower-middle-income and low-income countries and present ICT as one of the factors of economic development. On the other hand, the results show that ICT use is not significant for developed countries with high income levels. The results of the paper may serve as a recommendation in the creation of development strategies and policies both from the perspective of ICT development and implementation and from the perspective of human, i.e., overall, development of national economies. While ICT development in developing countries can serve as a tool for human and economic development, policies and actions in developed countries need to focus on other factors that can lead to higher levels of human development. Developing countries could integrate more easily into the current economic environment if they made the most of the opportunities offered by new technologies. The people living in these countries could also have easy access to new knowledge and information, health, job opportunities, etc.

The contribution of this paper also lies in the dynamic nature of the analysis in showing how human development in previous periods has a positive and significant impact on human development today. In this sense, the main objective of the study is to fill the gap in the literature by providing evidence of the dynamic nature of human development

and the link between ICT and human development, especially from the perspective of developing countries.

The research consists of six interconnected parts. After the introduction, the second part of the paper provides a review of the existing theoretical and empirical literature. The third part of the paper describes and defines the research methodology, while the fourth part describes the data and variables used in the model. The fifth part of the paper represents the research results, and the sixth part includes the discussion. The seventh part of the paper, the conclusion, synthesizes the paper.

## 2. Literature Review

The role of ICT in economic growth and development has long attracted great attention from economists, researchers, and policymakers. However, there is still no standard definition of ICT, but instead many definitions by different authors and institutions describing ICT from different perspectives. UNESCO (2021), for example, provides a broader ICT definition, arguing that ICT is "a diverse set of technological tools and resources used for the transmission, storage, creation, sharing or exchange of information. These technological tools and resources include computers, the Internet (websites, blogs, and e-mails), live broadcasting technologies (radio, television, and web broadcasting), recorded broadcasting technologies (podcasting, audio and video devices and storage devices), and telephony (fixed or mobile, satellite, visio/video conferences, etc.)". According to the OECD (2017), "ICT refers to different types of communication networks and technologies they use", while Sarkar (2012) claims that the term ICT refers to a diverse set of technological equipment and resources used for communication. Pradhan et al. (2018) claim that ICT infrastructure implies digital telephone networks, mobile telephones, Internet access, Internet servers, fixed broadband networks, and similar technologies. Therefore, as technology develops, new definitions of ICT emerge that further clarify and deepen the significance and importance of ICT.

In the past two decades, the empirical literature has been engaged mainly in the evaluation of the effects of ICT on economic growth (Pohjola 2001; Oulton 2002; Kuppusamy and Santhapparaj 2005; Bollou and Ngwenyama 2008; Zhang and Lee 2007; Kim et al. 2008; Farhadi et al. 2013; Toader et al. 2018), but the assessment results do not always show significant and positive effects. Gholami et al. (2010) argue that this is because the ICT effect is not always automatic, but depends, for example, on implementation in different sectors. It is therefore better to study the impact of ICT on education, the health system, human development, or the profitability of firms. Therefore, scholars have begun to relate ICT implementation to different variables that reflect one or more sides of human development, such as health, education, job creation, and overall quality of life. For example, Chege and Wang (2019) studied the impact of information technology on job creation in SMEs, and their results show that technological innovations positively influence job creation in small businesses and act as drivers of economic development. Das et al. (2020) support these findings and suggest that a good technological environment created by governments and institutions leads to SME development and job creation in developing countries.

Ngwenyama et al. (2006) analysed the effects of ICT investments on human development (through the prism of education and health), and the results of their research show a positive link between ICT and human development, i.e., the authors claim that joint investments in ICT, health, and education can significantly increase development. Gholami et al. (2010) investigated the link between ICT and human development and used the Human Development Index (hereinafter: HDI) as an indicator of human development. Their research results show that ICT has a more significant positive effect on human development in less developed countries than in highly developed countries. Bankole et al. (2011, 2013) assessed the effects of ICT investments on human development. In their paper, they differentiate ICT across four dimensions: hardware, software, research and development, and investments in telecommunications; human development is measured by GDP per capita, expected years of schooling, literacy level, and life expectancy. Research

results show the connection between ICT investments and human development, but ICT has a different impact on human development components, and these impacts are different in high-, middle- and low-income countries. Asongu and Le Roux (2017) investigated how the enhancement of ICT benefits human development in Sub-Saharan Africa, i.e., developing countries. Their results suggest that policies aimed at increasing the diffusion of ICT (mobile phone, internet, telephone) increase inclusive human development. In their research, De la Hoz-Rosales et al. (2019) analysed how the use of ICT by individuals, enterprises, and governments affects human development as measured by two indicators: the HDI and the Social Progress Index (SPI). The results of their analysis show that, regardless of the level of countries' development, individual ICT use has significant positive effects on human development, especially when measured by the HDI. Furthermore, in the case of ICT use in enterprises, the effects on human development are significant and positive at the global level, but are not significant when limited only to developed countries. On the other hand, the effects of government use of ICT are positive and significant for human development, also in the case of developed countries.

Balouza (2019), who investigated the effects of ICT on human development in six countries of the Gulf Cooperation Council (GCC), also had contrasting insights. According to the results of his analysis, ICT effects are inconsistent, i.e., they vary between positive, negative, and non-significant, and, in fact, show the real complexity of ICT implementation in certain countries. Balouza (2019) believes that while the GCC countries invested in ICT and the related infrastructure, they have not achieved the desired results due to discrepancies between technology and qualified human capital, adequacy of the educational system, and public awareness of the importance of ICT in socio-economic development. Namely, technology is insufficient if it is not adequately implemented. This is also backed by Amartya Sen's claim (1999) that the quality of human life depends on what people are capable of doing or what tools they have at their disposal. Sen (2010) also notes that it is important to identify how resources such as ICT can help people to be more efficient in their work and how their use can expand their capabilities.

Due to the contradictory results offered by the current empirical literature, this paper focuses on ICT effects on human development, taking into account the countries' level of development (measured by income per capita). The main question is, how do we measure human development at the macro-economic level? According to Karaman Aksentijević and Ježić (2017), "Human resources at the national level can be defined as the total mental and physical energy possessed by the inhabitants of a country, which cannot be expressed directly in value; therefore, the value and level of human development are measured indirectly." According to the UN, human development is defined as the development of the people, for the people, and by the people (HD Report 1998 in Karaman Aksentijević and Ježić 2009). A measure that is often used as an indicator of human development that also serves as an indicator of economic development, as it includes all important dimensions of human development, such as income, education, and health, is the Human Development Index (HDI). The authors must concede that the HDI does not fully reflect quality of life, as it does not capture the subjective side of human life, such as life satisfaction and self-esteem, which also affect the economic sphere (Kuzior and Kuzior 2020). However, it serves as a reasonable measure that is widely accepted in the empirical literature. "The HDI was constructed in the early 1990s by Amartya Sen (Nobel Laureate), Mahub ul Hak, Gustav Ranis (Yale University), Meghan Desai (London School of Economics)", and has been used since by the UN and published in the Human Development Report (Karaman Aksentijević and Ježić 2018). The HDI is calculated as a composite index that includes three indicators: 1. life expectancy and health of the population as measured by life expectancy; 2. knowledge and education of the population; 3. standard of living of the population as measured by GDP per capita (UNDP 2020). Ranis et al. (2006) believe that the indicators included in the HDI are insufficient and that index calculation should be extended by an even larger set of indicators. For the time being, however, the three-component HDI has

been used as an indicator of human and economic development. Therefore, the HDI is used as a dependent variable in this paper.

In general, the economic empirical literature agrees that ICT has significant and positive effects on human development, claiming that ICT has become the foundation for most social and economic progress in developed and developing countries (Selwyn 2004) and that ICT as a tool may help in the economic development of poor countries (Bankole et al. 2013). However, the results of individual surveys (Balouza 2019) show that this contribution remains disputable. Furthermore, on the microeconomic level, from a psychological perspective, ICT also negatively affects human life, as it causes technological stress (Salanova Marisa and Ventura 2014), anxiety (Kessler and Ustun 2008), and internet addiction (Porter and Kakabadse 2006; Douglas et al. 2008). These authors consider that ICT may even negatively affect the quality of human life. However, a strong belief exists among scholars that ICT implementation and adaption could benefit many aspects of human life and the economy, directly and/or indirectly, through access to information, knowledge, job opportunities, healthcare, internationalization, etc., in developing countries. Our study most closely fits into a growing stream of literature on the role of ICTs in economic and social development (Mir and Dangerfield 2013; Asongu 2015; Asongu and Le Roux 2017). This paper re-opens the question of ICT effects on human development from the macroeconomic point of view. The effects are assessed separately for countries of different levels of development to contribute to the current findings regarding the correlation between ICT and human development and fill the gap in the existing empirical literature.

## 3. Methodology

The aim of this paper is to evaluate the effects of information and communication technology on human development. The empirical analysis is based on dynamic panel data regression analysis. Panel data include a combination of timeline and time-series data. Since most economic variables have dynamic behaviour, i.e., the current value of a variable is dependent on previous values of the same variable, dynamic panel models are used in the analysis. By integrating lags of the dependent variable as explanatory variables, dynamic panel data models account for the dynamic nature of the interactions between economic variables. As a result, we employ Arellano and Bond's (1991) Generalized Method of Moments (GMM), which is one of the most extensively used approaches in empirical economic research. GMM estimators are also used in research related to human development, such as Asiama and Quartey (2009) and Akinbode and Bolarinwa (2020). Therefore, we found GMM specifically applicable to the present case.

The GMM estimator uses lags of endogenous variables as instruments to overcome endogeneity concerns caused by interactions between dependent and explanatory factors. In our case, the endogenous variable is the HDI (proxy for human development); we use lagged values of the HDI because human development is dynamic in nature. The GMM estimator delivers unbiased and consistent parameter estimates in this way (Greene 2008). Furthermore, when the number of periods T is small but the number of cross-sectional units N is large, as in our case, there are regressors that are not strictly exogenous (endogenous regressors), and fixed effects exist. The GMM approach is particularly well suited for estimating panel data. When heteroscedasticity and autocorrelation occur within country data but not across countries, it is also valuable.

To assess these effects, the following econometric model is estimated in the paper:

$$HDI_{it} = \beta_0 + \beta_1 HDI_{it-1} + \beta_2 GINI_{it} + \beta_3 ICTuse +_{it} \beta_4 Rule of Law_{it} + \lambda_t + u_{it} \quad (1)$$

where the Human Development Index (*HDI*) is the dependent variable, used as a proxy variable for human development in the country *i* in the year *t*, while independent variables are the lagged value of the dependent variable, namely the HDI of country *i* in time *t* − 1, the GINI coefficient (*GINI*), which serves as a proxy variable for development disparity, ICT use (*ICTuse*), and the rule of law as a proxy variable for institutions (*RuleofLaw*) in the country *i* in the year *t*. Finally, $\lambda_t$ presents yearly fixed effects and $u_{it}$ presents a model

error. The values of variables are expressed at country levels. However, to assess the effects on the HDI for each group of countries, we created four dummy variables for each group of countries: high, upper-middle, lower-middle, and low-income countries. The value of a dummy variable may be 0 or 1. In this model, if a country belongs to one of these groups, the variable value is 1, and if it does not, the variable value is 0. We estimated the model four times, separately for each group of countries, and the results of each regression are shown in Table with the results in columns (1)–(4).

## 4. Data and Description of Variables

The analysis was conducted on a database that includes 130 countries in the period 2007–2019. We extrapolated the GINI and ICTuse variables for 2018 and 2019 using linear extrapolation, i.e., the following formula:

$$y = \frac{y_1 - y_0}{x_1 - x_0}(x - x_0) + y_0 \tag{2}$$

where $y$ represents a variable and $x$ a year.

The analysis is based on country-level secondary macroeconomic data taken from different sources. Table 1 shows all the variables, i.e., indicators used in the model, a description of each indicator, and the source from which the indicators were taken. Our main variable of interest, human development, is measured with the Human Development Index (HDI). As presented in Table 1, HDI covers the area of life expectancy, education, and adequate standard of living, but it does not capture some subjective elements of the quality of human life, such as self-satisfaction, happiness, etc. However, it presents an adequate indicator of human development on a country level.

**Table 1.** Description of variables and sources.

| Variable | Indicator | Description | Source |
|---|---|---|---|
| Dependent variable | Human Development Index (HDI) | It measures average achievement in the three basic dimensions of human development—life expectancy, education, and adequate standard of living. | UN Human Development Report |
| Independent variables | GINI coefficient | The GINI coefficient is based on a comparison of cumulative shares of population with cumulative shares of income they receive, ranging between 0 in the case of perfect equality and 1 in the case of perfect inequality. | World Bank |
| | ICT use | ICT use includes several indicators: Internet users Broadband Internet subscriptions Internet bandwidth Mobile broadband subscriptions Mobile telephone subscriptions Fixed telephone lines | World Economic Forum, Global Competitiveness Report |
| | Rule of Law | The rule of law records the perception of the extent to which agents trust the rules of society and the rules of the society they adhere to, especially rules such as the quality of execution of contracts, property rights, police and courts, as well as the likelihood of crime and violence. | World Bank |

Source: Authors' work.

In order to group countries by income, we created four dummy variables for each group of countries using the new GNI Classification of the World Bank from 2020. The countries are grouped as follows:

- High-income countries with GNI/PC > USD 12,535;
- Upper-middle-income countries with GNI/PC > USD 4046 ≤ 12,535;
- Lower-middle-income countries with GNI/PC > USD 1036 ≤ 4045;
- Low-income countries with GNI/PC < USD 1036.

Table 2 shows descriptive statistics of all variables used in the model. The table shows that descriptive statistics are presented in total for all panel units, in our case countries, as well as between panel units and within a panel unit (variations of variables over the years for a single country).

**Table 2.** Descriptive statistics.

| Variables | | Average | Standard Deviation | Minimum | Maximum | Number of Observations |
|---|---|---|---|---|---|---|
| HDI | total | 0.7032662 | 0.1573503 | 0.306 | 0.957 | N = 2077 |
| | between | | 0.1562623 | 0.3540769 | 0.945 | n = 161 |
| | within | | 0.0201986 | 0.608497 | 0.7698816 | T-bar = 12.9006 |
| GINI | total | 38.27673 | 8.257535 | 19.87143 | 67.69999 | N = 1569 |
| | between | | 7.729933 | 24.70769 | 62.96154 | n = 145 |
| | within | | 2.656554 | 16.07673 | 60.47672 | T-bar = 10.8207 |
| ICTuse | total | 3.156548 | 1.823172 | −2.16215 | 9.783863 | N = 1876 |
| | between | | 1.539591 | 0.8493172 | 6.736362 | n = 148 |
| | within | | 0.9792761 | −1.814421 | 8.120159 | T-bar = 12.6757 |
| Rule of Law | total | −0.0577473 | 0.9788182 | −2.255286 | 2.100273 | N = 2287 |
| | between | | 0.972201 | −1.827445 | 1.991326 | n = 179 |
| | within | | 0.1427193 | −0.9053768 | 0.6724207 | T-bar = 12.7765 |

Source: Authors' calculation.

When observing standard deviation, i.e., the dispersion of the data relative to its mean, it is evident that for all variables it is greater between panel units (between countries) than within a panel unit, i.e., a country. This is logical, since it is based on countries at different levels of development, and there is a greater difference in the values of variables between countries over time than within a single country over time. A small variation of values is noticeable within panel units over time (2007–2019).

## 5. Results

The estimated results for GMM are presented in Table 3. In the model specifications, the Arellano and Bond two-step system GMM estimator was used. The results in the table are organized into four columns: the first column (1) shows the coefficients of the effects of independent variables on the dependent variable, HDI, in the case of high-income countries. The second column (2) shows the coefficients of the impacts of independent variables on the dependent variable, HDI, in the case of upper-middle-income countries. The third column (3) shows the impact coefficients in the case of lower-middle-income countries, while the fourth column (4) shows the coefficients for low-income countries.

**Table 3.** Results of the two-step system GMM estimation.

| | Dependent Variable—HDI | | | |
|---|---|---|---|---|
| | **(1)** | **(2)** | **(3)** | **(4)** |
| | **High-Income Countries** | **Upper-Middle-Income Countries** | **Lower-Middle-Income Countries** | **Low-Income Countries** |
| **Variables** | **HDI** | **HDI** | **HDI** | **HDI** |
| L.HDI | 0.847 *** | 0 | 0.878 *** | 0 |
| | (0.106) | (0) | (0.0380) | (0) |
| GINI | −0.000525 | −0.00238 | −0.000189 | −0.00755 *** |
| | (0.000469) | (0.434) | (0.000214) | (0.000359) |
| ICTuse | −0.000134 | 0.00730 | 0.00566 * | 0.0537 *** |
| | (0.000770) | (4.392) | (0.00302) | (0.00675) |
| RuleofLaw | 0.0124 * | −0.0234 | 0.0163 *** | −0.141 *** |
| | (0.00700) | (9.756) | (0.00600) | (0.00446) |
| Time fixed effects | YES | YES | YES | YES |
| Constant | 0.140 * | 0.816 | 0.0867 *** | 0.590 *** |
| | (0.0845) | (23.80) | (0.0225) | (0.00832) |
| Observations | 434 | 339 | 270 | 136 |
| Number of panels | 38 | 31 | 28 | 15 |
| Hansen statistics | 23.30 | 270.28 | 5.40 | 0 |
| Hansen *p*-value | 0.950 | 0.000 | 1 | 1 |

Robust standard errors in parentheses. *** $p < 0.01$, ** $p < 0.05$, * $p < 0.1$. Source: Authors' calculation.

The two-step system GMM estimation results show that the lagged value of HDI has highly significant and positive effects on HDI in the present period, specifically in case of high-income and lower-middle-income countries. In case of upper-middle-income countries and low-income countries, a lagged value of HDI was omitted from the estimation, because yearly effects are correlated with the variable (HDI lag). The impact of the GINI coefficient is negative in all estimates, but significant only in the case of low-income countries. The negative sign is in line with expectations, given that the higher the value of the GINI coefficient, the greater the inequality. The variable we are the most focused on, ICT use, shows strong positive and significant ($p < 0.01$) effects on HDI in the case of low-income countries with a GNI/PC < USD 1036 and in the case of low- and middle-income countries with a GNI/PC > USD 1036 ≤ 4045. However, our results suggest that ICT does not appear to be significant in the case of upper-middle-income countries with a GNI/PC > USD 4046 ≤ 12,535 and in the case of high-income countries with a GNI/PC > USD 12,535. The table also shows that most of the countries in the sample, 38, are high-income countries, while the number of low-income countries, 15, is the smallest. In the case of the rule of law variable, the proxy variable for institutions, the results are quite ambiguous. Namely, the impact of institutions on human development is positive and significant in developed countries, i.e., high-income countries, while it is not significant in middle- and low-income countries. In contrast, institutions have a significant but negative impact on human development in the case of low- and middle-income countries.

## 6. Discussion

From these results, we can conclude that ICT is important for human development in less developed countries, i.e., lower income countries. It is interesting to observe that the less developed the country, the greater the impact of ICT on human development. Indeed, developed countries have already reached a high level of development and have much less room for development through ICT implementation. The high usage of ICT in developed countries might even cause some negative effects on human development related to ICT effects on human health. Less developed countries have much more room for development through ICT implementation. These countries have invested in and implemented less ICT

so far, and with greater investment and implementation, they can make much larger shifts and achieve higher levels of development. Regarding the institution variable (rule of law), we can assume that institutions in high- and upper-middle-income countries are regulated, and therefore their impact on human development is significant, while institutions in lower-middle-income countries are still in the process of development and regulation, so this could explain why their impact is negative and opposite to our expectations. Low-income countries have not yet reached the level of institutional development that would result in a positive impact on human development, i.e., HDI. It is also possible that institutional development in the least developed countries itself contributes little to human development compared to other measures.

When we compare our results with the literature, we see that our results support previous research (Gholami et al. 2010; Bankole et al. 2011, 2013; De la Hoz-Rosales et al. 2019) in which the effects of ICT on HDI were confirmed, with more significant effects in developing countries than in highly developed countries, i.e., high-income countries. The reason could be that highly developed countries have reached a certain peak in ICT development and that technological innovation such as artificial intelligence and cognitive technologies (see Kwilinski et al. 2019) could be more important for them than ICT use per se.

## 7. Conclusions

The aim of this paper was to examine the impact of information and communication technology (ICT) on human development by controlling for other variables that might have an impact on human development. Human development was measured using the Human Development Index (HDI), as this indicator captures life expectancy and thus indirectly population health, knowledge and education, and standard of living and has been proposed in the literature as a good measure of human development. In addition, the aim of our research was to investigate whether ICT has differential impacts on human development depending on the level of income, i.e., the level of development of countries. Thus, we wanted to investigate whether ICT is equally important for countries at different levels of development, and if so, in which countries it has a stronger impact. This information is extremely important from the perspective of policymakers, especially in developing countries that could benefit from investment in ICT. Moreover, the dynamic nature of human development is often neglected in the empirical literature, so in this paper we also wanted to capture the impact of human development in earlier periods on human development today.

The results of our research support the dynamics of human development and show that the HDI has statistically significant and positive effects on contemporary human development, which is noteworthy because the same countries usually perform best or worst. Regarding the impact of ICT, the estimation results are broadly in line with expectations and show that ICT is important for human development and has a positive impact on the HDI in the case of developing countries (middle- and low-income countries). In the case of developed countries, on the other hand, the results are ambiguous and not significant. When a country reaches a higher level of development, i.e., higher income level, these effects are smaller or not significant in the case of high-income countries. ICT seems to be important for developing countries, which means that development policies and strategies should consider investments in ICT as important. This paper contributes to the recognition of ICT as one of the determinants of human development, and hence economic growth and development, and supports previous findings on the subject. However, this paper also contributes to the existing body of knowledge by clearly demonstrating the impact of ICT on human development in countries at different levels of development. Recognizing the importance of ICT for human development is extremely important for less developed countries, as our results support the claims of the empirical literature that ICT has significant effects for less developed countries. At the macroeconomic (political) level, there is scope for improving governments and institutions and the effectiveness of their

actions, with the aim of reducing the barriers that currently prevent developing countries from benefiting directly from ICT opportunities. In addition, this paper contributes to the existing literature by showing the relationship between human development in different time periods. These results can serve as a reference in development policy making and help fill the gap in the literature by providing insights into the relationship between information and communication technology and human development.

The main limitations of this research are reflected in the use of aggregated data, where some information is sometimes lost through the aggregation itself. We must acknowledge that our dependent variable, the HDI, does not encompass all aspects of human development and the quality of people's lives, especially subjective indicators. Future research could be extended by including other indicators that could better reflect the quality of human life. Our results also show that ICT use per se is insignificant for human development in high- and middle-income countries, which raises new questions about which technologies might promote human development in these countries. Indicators such as the development of artificial intelligence and cognitive technologies (mentioned in Kwilinski et al. 2019) could provide new insights into the relationship between technology and human development.

In conclusion, although we aimed to study ICT effects at the macroeconomic level, the research can be further extended and deepened by analysing the effects of ICT use at the regional level, where the model would include variables characteristic of individual regions, such as cultural aspects. For future research, the HDI calculation methodology could also be applied at the regional level to provide a more detailed insight into human resource development at the regional level. This would contribute to the existing theory of regional growth and development.

**Author Contributions:** Conceptualization, N.K.A., Z.J. and P.A.Z.; methodology, Z.J. and P.A.Z.; software, P.A.Z.; validation, N.K.A., Z.J. and P.A.Z.; formal analysis, N.K.A., Z.J. and P.A.Z.; investigation, N.K.A., Z.J. and P.A.Z.; resources, Z.J. and P.A.Z.; data curation, P.A.Z.; writing—original draft preparation, N.K.A., Z.J. and P.A.Z.; writing—review and editing, N.K.A., Z.J. and P.A.Z.; visualization, N.K.A. and P.A.Z.; supervision, N.K.A. and Z.J.; project administration, Z.J.; funding acquisition, Z.J. All authors have read and agreed to the published version of the manuscript.

**Funding:** This paper was funded under the project line ZIP UNIRI of the University of Rijeka, for the project ZIP-UNIRI-130-9-20 (E-)education and Human Resources Development.

**Data Availability Statement:** Data used in this article is freely available at http://hdr.undp.org /en/content/human-development-index-hdi; https://data.worldbank.org/indicator/SI.POV.GI NI; https://tcdata360.worldbank.org/indicators/gci?country=BRA&indicator=631&viz=line_char t&years=2007,2017; https://datacatalog.worldbank.org/dataset/worldwide-governance-indicators [accessed on 10 February 2021].

**Conflicts of Interest:** The authors declare no conflict of interest.

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
