# Peer review of "The Effects of Information and Communication Technology (ICT) Use on Human Development—A Macroeconomic Approach"

_economies, doi:10.3390/economies9030128_

Round 1

Reviewer 1 Report

Dear Author/Authors

The article entitled "The Effects of Information and Communication Technology (ICT) Use on Human Development - the Macroeconomic Approach" deals with the current issues of the impact of ICT on human development. One should agree with the Author / Authors that ICT has a significant impact on growth and development (especially in developing countries), but in the opinion of the reviewer, there is no direct cause-effect relationship between the impact of ICT, growth and development on a better quality of life. The Author/Authors should verify their position in this regard and possibly support it with previous research or their own research confirming such a direct relationship. The research of psychologists shows that ICT also has negative effects and, as a result, reduces the quality of human life, which indirectly translates into the economic sphere.

The abstract of the article is basically complete and reflects the essence of the entirety of the research carried out and described. It highlights the significant positive impact of ICT on human development in developing countries. The Author/Authors, however, do not mention developed countries, although such analyzes are carried out in the further part of their scientific argument. I suggest that the summary should be completed with this aspect. This may interest the reader, for whom such a comparison will probably be interesting and encourage him to read the entire article.

In introduction the Author/Authors introduce the term "new normal". It is written in quotation marks and requires an explanation of what the Author/Authors mean by this term.

The part of literature review is quite one-sided. It does not take into account those studies that indicate the negative impact of ICT on human development. There are many scientific studies on the impact of ICT on human development, taking into account various dimensions of its functioning. I suggest extending this part and at least mentioning also these negative impacts of ICT on human development.

Moreover, the Author/Authors only pay attention to objective indicators of the quality of life, such as income, health, education, and never mention subjective indicators of the quality of life, such as life satisfaction and self-esteem, which also affect the economic sphere, e.g. efficiency at work, competences of the so-called society 4.0 (cf. Kuzior, A., & Kuzior, P. (2020). The Quadruple Helix Model as a Smart City Design Principle.Virtual Economics, 3(1), 39-57. https://doi.org/10.34021/ve.2020.03.01(2) ). It would be enough to mention that these threads are important.

I leave the Author/Authors to consider the issue of the development of artificial intelligence and cognitive technologies that have a significant impact on social development and ICT (cf. Kwilinski, A., Tkachenko, V., Kuzior, A. (2019). Transparent cognitive technologies to ensure sustainable society development. Journal of Security and Sustainability Issues, 9(2), 561-570 http://doi.org/10.9770/jssi.2019.9.2(15))

Methodology could be improved.

In conclusion, the Author / Authors indicate that the results of their research provide new insights and can be used to shape development policies. Meanwhile, the described results are only a confirmation of the results of the previously conducted research, as the Author/Authors themselves point out. The Author/Authors at this point need to highlight these new insights resulting from their research.

Good luck with revisions and I look forward to a response.

Author Response

Dear Reviewer,

we would like to thank you for taking the time to read our paper, for your valuable comments, and for the opportunity to resubmit the paper. We believe we have revised all the issues outlined below.

Please find below a list of detailed comments in response to reviewer’s suggestions. The changes and additions are set as trach changes as proposed by the Journal.

We remain at your disposal for any additional modifications you may require.

Sincerely,

Authors

1. The article entitled "The Effects of Information and Communication Technology (ICT) Use on Human Development - the Macroeconomic Approach" deals with the current issues of the impact of ICT on human development. One should agree with the Author / Authors that ICT has a significant impact on growth and development (especially in developing countries), but in the opinion of the reviewer, there is no direct cause-effect relationship between the impact of ICT, growth and development on a better quality of life. The Author/Authors should verify their position in this regard and possibly support it with previous research or their own research confirming such a direct relationship. The research of psychologists shows that ICT also has negative effects and, as a result, reduces the quality of human life, which indirectly translates into the economic sphere.

We agree that there is an indirect relationship between ICT and better quality of life, but the choice of variables in the empirical model is based on previous findings mentioned in the literature review such as Gholami et al. (2010), Bankole et al. (2011, 2013), de la Rosales et al. (2019). In our paper, due to data limitations related to the human’s quality of life, we use HDI as proxy variable that does not cover all aspects of the quality of humans life (which we have acknowledged in paper) but is widely accepted and used in the empirical literature. We have explained our model specifications and choice of variables in more detail in the paper. The connection between the impact of ICT, growth and development on a quality of life is part of multiple papers (including textbook of the authors of the paper). Namely, as stated in paper, there is a direct connection between technology development and economic development. Quality of life has direct link with the economic development. Quality of life, according to different authors, can be measured with different indicators and most of them imply availability of technology in mass usage. One of the interesting papers was published in June 2016 on the 1st International conference on Quality of Life entitled: Impact of ICT on Quality of Life (http://cqm.rs/2016/cd1/pdf/papers/focus_1/32.pdf). In addition, we also mentioned the negative effects of ICT, as recommended by the reviewers.

2. The abstract of the article is basically complete and reflects the essence of the entirety of the research carried out and described. It highlights the significant positive impact of ICT on human development in developing countries. The Author/Authors, however, do not mention developed countries, although such analyses are carried out in the further part of their scientific argument. I suggest that the summary should be completed with this aspect. This may interest the reader, for whom such a comparison will probably be interesting and encourage him to read the entire article.

We agree with the comments and have included the results in the summary in the case of developed countries.

3. In introduction the Author/Authors introduce the term "new normal". It is written in quotation marks and requires an explanation of what the Author/Authors mean by this term.

The "new normal" is a term for the state in which an economy, society, etc., finds itself after a large crisis or shifts (especially in technology), when it differs from the situation that prevailed before the onset of the crisis. Last large impact to “new normal” or “new average” happened because of COVID-19 but according to the (for example) Forbes (https://www.forbes.com/sites/forbestechcouncil/2021/01/25/the-new-normal-and-the-future-of-technology-after-the-covid-19-pandemic/?sh=dfe9e256bbb3) and World Economic Forum (https://www.weforum.org/agenda/2020/06/theres-nothing-new-about-this-new-normal-heres-why/) the term New normal was used also post World War, 1990 dot com bubble, 2008 financial crisis…

Once again, thank you for the suggestion, we added publication Toward the “New Normal” After Covid-19 – a Post Transition Economy Perspective (https://www.researchgate.net/profile/Ewa-Minska-Struzik/publication/352707874_Toward_the_new_normal_after_Covid-19_-_a_post-transition_economy_perspective/links/60d46b2292851c8f79980c60/Toward-the-new-normal-after-Covid-19-a-post-transition-economy-perspective.pdf) and Digital Economy as a New Driver for Growth (https://link.springer.com/chapter/10.1007/978-981-33-6005-1_2  ) to the References.

4. The part of literature review is quite one-sided. It does not take into account those studies that indicate the negative impact of ICT on human development. There are many scientific studies on the impact of ICT on human development, taking into account various dimensions of its functioning. I suggest extending this part and at least mentioning also these negative impacts of ICT on human development.

We agree with the comment and have included the references that reflect the other side of the story, i.e., the negative effects as well.

5. Moreover, the Author/Authors only pay attention to objective indicators of the quality of life, such as income, health, education, and never mention subjective indicators of the quality of life, such as life satisfaction and self-esteem, which also affect the economic sphere, e.g. efficiency at work, competences of the so-called society 4.0 (cf. Kuzior, A., & Kuzior, P. (2020). The Quadruple Helix Model as a Smart City Design Principle.Virtual Economics, 3(1), 39-57. https://doi.org/10.34021/ve.2020.03.01(2) ). It would be enough to mention that these threads are important.

Our scope is narrower, but we agree with the reviewer's suggestions and have included the above references in our literature review. Namely, we use the HDI as a proxy variable for human development, although we are aware that the HDI does not cover all aspects of people's quality of life. We have acknowledged this limitation, which can also be a stimulus for further research.

6. I leave the Author/Authors to consider the issue of the development of artificial intelligence and cognitive technologies that have a significant impact on social development and ICT (cf. Kwilinski, A., Tkachenko, V., Kuzior, A. (2019). Transparent cognitive technologies to ensure sustainable society development. Journal of Security and Sustainability Issues, 9(2), 561-570 http://doi.org/10.9770/jssi.2019.9.2(15)).

We believe that this topic is too broad for the aim of this research, but it was worth considering, so we have mentioned it in the proposals for the extension of this research. Also, we delt with the topic of new technologies (AI, deep learning, cloud computing, clouting, virtual and augmented reality) and connection with economic development in our two books (handbook Economics of Development and scientific monograph Human Resources and Economic Development) and other papers. In this paper, because we wanted to explore the connection between ICT and economic development on the Macroeconomic approach, we used sample of 130 countries and available data about ICT in those countries. The authors believe that in the future international statistics will cover the data about teraflops of processing power and it will be possible to establish direct connection between AI and growth and development of a country.

7. Methodology could be improved.

As the reviewer did not suggest to what extent the methodology could be improved, we have better explained the model specifications used to conduct the study and added additional scientific papers to our research.

In conclusion, the Author / Authors indicate that the results of their research provide new insights and can be used to shape development policies. Meanwhile, the described results are only a confirmation of the results of the previously conducted research, as the Author/Authors themselves point out. The Author/Authors at this point need to highlight these new insights resulting from their research.

We have done a better job of highlighting the new findings that emerge from our research.

Thank you once again for all the insights and great suggestions. Hopefully we have been able to answer all of them.

Reviewer 2 Report

In my opinion, the article deals with an interesting and important topic. However, the effects of Information and Communication Technology (ICT) use on human development from a macroeconomic perspective has already been conducted in many papers. Through this study, additional explanations are needed on how to provide policy implications, especially for low-income countries.

A good structure of the text is an asset.  

Here are some comments to improve the paper:

[1] Research framework and hypothesis development

The research methodology seems to have been well done. There is a problem with the hypothesis-setting process. It is necessary to present clear your research hypothesis and relevant research in the hypothesis hypothesis-setting process.

[2] Conclusions

The conclusions should be extended, the conclusion is minor in relation to the scope of the research. Authors must elaborate more on what is their contribution to the literature as well as an opportunity for future research. Questions that you need to answer: (1) Why is your study important? (2) How could it extend to the existing knowledge on the issue/topic? I suggest you concentrate on the description of the implications of your main findings.

 Good luck!

Author Response

Dear Reviewer,

we would like to thank you for taking the time to read our paper, for your valuable comments, and for the opportunity to resubmit the paper. We believe we have revised all the issues outlined below.

Please find below a list of detailed comments in response to reviewer’s suggestions. The changes and additions are set as trach changes as proposed by the Journal.

We remain at your disposal for any additional modifications you may require.

Sincerely,

Authors

In my opinion, the article deals with an interesting and important topic. However, the effects of Information and Communication Technology (ICT) use on human development from a macroeconomic perspective has already been conducted in many papers. Through this study, additional explanations are needed on how to provide policy implications, especially for low-income countries.

We agree with the comments and have better explained and expanded the policy implications of our research, focusing particularly on low-income countries.

A good structure of the text is an asset.  

Here are some comments to improve the paper:

[1] Research framework and hypothesis development

The research methodology seems to have been well done. There is a problem with the hypothesis-setting process. It is necessary to present clear your research hypothesis and relevant research in the hypothesis hypothesis-setting process.

We have made the research hypothesis more explicit and highlighted in the literature review the relevant research findings that support our hypothesis.

[2] Conclusions

The conclusions should be extended, the conclusion is minor in relation to the scope of the research. Authors must elaborate more on what is their contribution to the literature as well as an opportunity for future research. Questions that you need to answer: (1) Why is your study important? (2) How could it extend to the existing knowledge on the issue/topic? I suggest you concentrate on the description of the implications of your main findings.

We have expanded the conclusion in line with the reviewers' suggestions, which means that we have highlighted the importance of this research and its contribution to the existing body of knowledge.

Thank you once again for all the insights and great suggestions. Hopefully we have been able to answer all of them.

Round 2

Reviewer 1 Report

Dear Author / Authors
Thank you for your good work and the corrections in this article.
I accept it in the current form.
I wish you continued good research work.
Kind regards
Reviewer

Author Response

Dear Reviewer,

thank you for your time and comments, which have made possible a better version of our work.
Kind regards,
Authors

This manuscript is a resubmission of an earlier submission. The following is a list of the peer review reports and author responses from that submission.